# Fabrication of Copper of Harmonic Structure: Mechanical Property-Based Optimization of the Milling Parameters and Fracture Mechanism

**DOI:** 10.3390/ma15238628

**Published:** 2022-12-03

**Authors:** Phanumas Sojithamporn, Choncharoen Sawangrat, Komgrit Leksakul, Bhupendra Sharma, Kei Ameyama

**Affiliations:** 1Department of Industrial Engineering, Faculty of Engineering, Chiang Mai University, Chiang Mai 50200, Thailand; 2Advanced Manufacturing and Management Technology Research Center (AM2Tech), Department of Industrial Engineering, Faculty of Engineering, Chiang Mai University, Chiang Mai 50200, Thailand; 3Department of Mechanical Engineering, Faculty of Engineering, Kyushu University, Fukuoka 819-0395, Japan; 4Department of Mechanical Engineering, College of Science and Engineering, Ritsumeikan University, Kyoto 525-8577, Japan

**Keywords:** mechanical milling, fracture mechanism, pure copper, bimodal grain structure, harmonic structure, mechanical property, optimization

## Abstract

A severe plastic deformation process for the achievement of favorable mechanical properties for metallic powder is mechanical milling. However, to obtain the highest productivity while maintaining reasonable manufacturing costs, the process parameters must be optimized to achieve the best mechanical properties. This study involved the use of response surface methodology to optimize the mechanical milling process parameters of harmonic-structure pure Cu. Certain critical parameters that affect the properties and fracture mechanisms of harmonic-structure pure Cu were investigated and are discussed in detail. The Box–Behnken design was used to design the experiments to determine the correlation between the process parameters and mechanical properties. The results show that the parameters (rotation speed, mechanical milling time, and powder-to-ball ratio) affect the microstructure characteristics and influence the mechanical performance, including the fracture mechanisms of harmonic-structure pure Cu specimens. The best combination values of the ultimate tensile strength (UTS) and elongation were found to be 272 MPa and 46.85%, respectively. This combination of properties can be achieved by applying an optimum set of process parameters: a rotation speed of 200 rpm; mechanical milling time of 17.78 h; and powder-to-ball ratio of 0.065. The superior UTS and elongation of the harmonic-structure pure Cu were found to be related to the delay of void and crack initiation in the core and shell interface regions, which in turn were controlled by the degree of strength variation between these regions.

## 1. Introduction

Metallic materials are widely used in the manufacturing industry and for structural applications due to their good ductility and high strength. However, developing high strength and ductility of metallic material simultaneously has been the great challenge for materials scientists [1,2]. Ultrafine-grained (UFG) metallic materials, which have extremely high strength compared with their coarse-grained (CG) counterparts, have generated considerable attention [1,3]. However, the limitation of UFG materials is their poor ductility, something which is the main barrier for widely used commercial applications [4]. Over recent years, modal grain size distribution has been shown to achieve nanoscale microstructure control via grain engineering to create ultrafine-grained homo- and heterostructures in metallic materials, with several having been studied [5,6,7,8,9,10].

Several fabrication methods have been introduced to offer various heterogeneous structured materials with balanced strength and ductility, such as the gradient structure produced by surface mechanical attrition [11,12,13], the hetero-lamellar structure produced by cold rolling and annealing [2,14,15], and the harmonic structure produced by powder metallurgy [16,17,18,19,20]. Mechanical milling, a metallurgy technique, has been applied for microstructure modification [21,22]. Mechanical milling under soft energy conditions has resulted in intensive plastic deformation of the outer part, by which one can observe the significant refinement of the microstructure, while the inner part remains unchanged [23]. Additionally, MM is used to create a peculiar bimodal grain structure in the milled powder, followed by spark plasma sintering or hot isostatic pressing to fabricate (harmonic structure) HS materials [24,25]. Harmonic structure as a novel tri-dimensional heterogeneous structure, proposed by Ameyama et al. [26] can be a good candidate that offers both strength and ductility in the same material [27,28,29,30,31]. Various historical studies have indicated that harmonic structures lead to the enhancement of mechanical properties [32]. Additionally, harmonic structures (HS) are constituted in the form of a three-dimensional, continuously connected network (‘‘shell’’) of fine-grained areas, and the areas with coarse-grained structure (‘‘core’’) remain present as islands embedded in the matrix. These specific characteristics lead to a combination of high strength and elongation [30,33]. MM is effective at promoting excellent mechanical performance [34] and is applicable to many industrial sectors. Furthermore, this process has been widely studied since it has potential for green, reproducible, low-cost, and large-scale production [35]. In the manufacturing process, it has been a great challenge to control and optimize process parameters to ensure reasonable manufacturing time and costs. Nevertheless, several papers have reported that MM is a very complex process that involves the optimization of many variables to achieve the desired mechanical properties. Such variables include the mechanical milling time, milling speed, powder-to-ball ratio, milling ball size, process control agent, milling atmosphere, milling agent, and starting powder size [36,37]. These process parameters influence both the microstructure and mechanical properties of the materials [38,39,40,41,42]. The correlation between the process parameters, microstructural characteristics, and mechanical properties has been highlighted in several studies. The mechanical property, the hardness, of CoCrFeMnNi high entropy alloys (HEA) has been shown to be improved after milling and sintering [43]. Similarly, Ghasemi et al. [44] have indicated that the hardness of mixtures of magnesium powder and SiC nanoparticles was enhanced. In contrast, the reduction of elastic modulus was observed after the mechanical milling process. The effect of atmosphere and milling time on the mechanical properties of Cu powders was presented by Madavali et al. [45]. Penther et al. [46] investigated the effect of the mechanical milling process on the microstructure and hardness of Mg–SiC nanocomposites. Their results demonstrate that the ultrafine grain of nanocomposite was achieved by the mechanical milling process which led to a strengthening of the composite. The microstructure and superconducting characteristics of the Y_3_Ba_5_Cu_8_O_y_ compound were studied by Slimani [47]. Their results indicate that the ball-powder weigh ratio 5:2 and speed rotation of 600 rpm generated a fine microstructure. The parameters of the mechanical milling process, including ball size, milling speed, and milling time, have been optimized in order to achieve the optimum particle size reduction [48]. The optimal milling time was 15 h, after which further milling time will not significantly enhance the mechanical properties. Neves et al. [49] studied the effect of MM process parameters on the enthalpy of a high-temperature reaction. Cu not only has good performance in conductivity and resistivity, but also has extremely rich reserves on Earth [50]. Therefore, extensive work related to the process parameters of MM to the microstructural characteristics of HS in pure Cu was presented by Sawangrat et al. [51]. In addition, the desired microstructure can be achieved by developing thermomechanical processing (TMP). Li et al. [52] showed that the micro-structural evolution and grain size distribution were characterized during thermos mechanical processing. The increased volume fraction of Shell, which lead to further improvement in the mechanical properties, occurred after TMP [53].

Bi-modal milling (BiM), a novel systematic fabrication approach, was proposed to prepare harmonic structures (HS). This technique precisely controls the microstructural features of the shell and core [54]. However, no broad studies have been conducted to im-prove the MM process by optimizing the process parameters for the mechanical properties and clearly understanding the correlation between the process variables and fracture mechanism in HS pure Cu. Consequently, it is important to extend research activities to achieve further understanding and enhancement of HS pure Cu. The scope of this work is to improve the MM process by optimizing the process parameters to obtain the best combination of ultimate tensile strength (UTS) and elongation, and to correlate these parameters with the fracture mechanism. The response surface methodology (RSM), broadly used for the method of process optimization [55] and employing the Box–Behnken design (BBD), was conducted to determine the statistical correlation of the process parameters and a second-order quadratic model was used to forecast the best combination of UTS and elongation. Honary et al. [56] have indicated that this method can reduce time and cost, and also provides precise results in practice. Moreover, RSM is suitable for evaluating multiple variables and their interactions since the number of experiments is reduced [57]. Subsequently, the harmonic microstructural evolution and microstructural features were investigated using electron backscatter diffraction (EBSD). The optimum set of process parameters was validated by comparing the experimental results. Furthermore, EBSD analyses were performed on the fracture surfaces of the tensile specimens to highlight the main fracture mechanism. Finally, the relationship between the optimum set of process parameters and the fracture mechanism of HM pure Cu were determined.

## 2. Material and Experimental Procedures

### 2.1. Sample Preparations

In this study, commercially pure Cu with an average particle diameter of approximately 160 µm was used. The initial powder was mechanically milled under an argon atmosphere at room temperature using a planetary ball mill with a tungsten carbide vial and SUJ2 steel balls. The milled powders were consolidated by spark plasma sintering in a vacuum atmosphere at 873 K with an applied pressure of 100 MPa for 1 h.

### 2.2. Microstructure and Mechanical Properties Characterization

The microstructural characteristics of the sintered compacted specimen, including the fracture surface, were examined by scanning electron microscopy (SEM) (JSM-IT300, JEOL., Ltd., Tokyo, Japan) with EBSD. The volume fraction of the UFG regions was estimated using digital image analysis of the microstructure micrographs. The tensile tests were carried out using a universal testing machine (AGS-10kND, Shimadzu Scientific Instruments, Inc., Columbia, SC, USA) at an initial strain rate of 5.6 × 10^−4^ s^−1^ (a constant crosshead speed of 0.1 mm/min) at room temperature. A non-contact extensometer based on CCD camera system was used to measure the strain during tensile tests. Tensile specimens with gauge lengths of 3 mm, widths of 1 mm, and thicknesses of 1 mm were prepared by electric discharge machining.

### 2.3. Identifying Potential Process Parameters and Their Possible Range

Historical literature, incorporated with a preliminary study from Sawangrat et al. [51], was investigated to explore the potential scope of the process parameters in the mechanical milling process, which in turn determine the microstructural characteristics and mechanical properties of pure Cu. From the investigation, the rotation speed (A), mechanical milling time (B), and powder-to-ball ratio (C) were selected for this study. The feasible limits of each process parameter were investigated based on the unique characteristics of the harmonic structure materials and the limitations of the experiment. The chosen levels and notations are shown in Table 1. With three experimental factors, each with three levels, an RSM using the Box–Behnken experimental design was performed on Minitab 16.0 to plan the experiment in this study (Table 2). The results of the Box–Behnken design were analyzed using ANOVA with a 95% confidence interval.

### 2.4. Development of Numerical Relationship

The RSM was applied to forecast the maximum UTS and elongation of pure Cu materials. The UTS and elongation were functions of the rotation speed (A), mechanical milling time (B), and powder-to-ball ratio (C) and could be expressed as
Ultimate tensile strength (UTS) = f (A, B, C),(1)
Elongation = f (A, B, C)(2)

## 3. Results and Discussion

### 3.1. Influence of Microstructure Character and Mechanical Performance

It is well-known that the microstructural characteristics of a material are strongly related to its mechanical performance. To determine the nature of these relationships, three experimental conditions presented in Table 2 were sampled, and the correlation between the microstructural features and mechanical performance was investigated. These three conditions were the same as in experiment #1 (A = 100 rpm, B = 10 h, and C = 0.045; hereafter called Sample I), #13 (A = 150 rpm, B = 15 h, and C = 0.045; hereafter called sample II), and #19 (A = 200 rpm, B = 20 h, and C = 0.045; hereafter called sample III), as listed in Table 2. Parameters A and B were the largest in sample III and smallest in sample I. Therefore, samples III, II, and I (experiments #19, 13, and 1, respectively) are referred to as having the maximum, middle, and minimum process parameter levels, respectively.

Figure 1a shows the microstructure of the base material. It can be clearly observed that the base material compacts have a mostly homogeneous coarse-grained (CG) microstructure with an average grain size of approximately 33.4 μm, as shown in Table 3. Figure 1b–d show representative microstructures of sintered compacts created from the minimum, middle, and maximum process parameter levels, respectively. It is clearly seen that sintered compacts with peculiar heterogeneity consist of UFG (darker contrast) and CG regions (brighter contrast). Sawangrat, et al. [51] have indicated that the darker contrast bands which contain high density grain boundaries bordering on a UFG microstructure increased with longer MM processing time. In particular, both regions have a unique topological distribution, where the region with UFG forms a 3-dimensional high-strength network, called a “shell,” which surrounds the isolated softer CG region, called a “core.” There are two regions of topological distribution including the region with severe deformation and an un-deformed region which are referred to as “shell” and “core”, respectively [58]. Such peculiar microstructural features are called “harmonic structures”. It is suggested that the aforementioned process parameter levels can be successfully used to prepare HS pure Cu. However, the mechanical properties of the specimen are expected to be different because of the differences in the microstructural characteristics induced by the different process parameter levels.

According to Table 3, the minimum process parameter levels have the largest mean core and shell grain size (30.12 µm and 4.3 µm, respectively) and result in the lowest UTS and elongation (231.03 MPa and 32.94%, respectively) when compared to the other processing conditions (excluding base material). The maximum process parameter level has the smallest core and shell mean grain size (25.87 µm and 1.8 µm, respectively) and leads to the highest UTS (282.29 MPa). However, the largest elongation (45.22%) is observed in the middle process levels. In addition, the shell volume fraction increases from 12.42% to 42.84% with increasing process parameter values. It can be clearly seen that the mechanical performance of the HS pure Cu is strongly dependent on the microstructural features, which are induced by the process parameter levels in the mechanical milling step. The process parameters in the mechanical milling step play an essential role in controlling the mechanical properties of HS pure Cu. Therefore, understanding these correlations was important in improving the stability of the mechanical milling process.

### 3.2. Influence of Process Parameters on Mechanical Performance

Table 4 shows the results of the Box–Behnken design. There is a total of 30 experiment runs with two replicates. According to Table 2, each experiment number is randomly assigned a “run” number to determine the order they would be performed in, to avoid bias in the experiment results due to the influence of some unknown factor. The experiment run numbers are also presented in Table 4, together with the values of the two response parameters.

Figure 2 shows the relationship between UTS and elongation as a function of each process parameter level. It is evident that UTS increases monotonically with an increase in the rotation speed and mechanical milling time over the range tested. According to the results from Corrochano [59], the trend of ultimate tensile strength correlates to the results in this research. In contrast, the elongation shows non-monotonic behavior. Over the selected sample points, the elongation grows at the middle parameter level, and decreases at the maximum parameter level. Similarly, Albaaji et al. [60] have demonstrated that the elongation of FeCo alloy was significantly increased during the first 6 h ball-milling but decreased beyond that point. It was suspected that the improved UTS would also reduce the elongation due to reducing the grain diameter and increasing the shell volume fraction caused by the increase in the process parameter values. These results indicate the complex relationship between the process parameters and both UTS and elongation, including the fact that the influence of the process parameter on each of the mechanical properties was not clear. Therefore, it was extremely important to concentrate the research effort towards developing an optimum set of process parameters and to determine the best combination of UTS and elongation.

Table 5 shows the results of the analysis of variance (ANOVA) based on the Box–Behnken design experiment for ultimate tensile strength (UTS) with all process parameters and their respective values, previously shown in Table 3. It can be clearly observed that the model is significant (*p* < 0.05). Moreover, it can be seen that the statistically significant parameters that affect the UTS (*p* < 0.05) are the rotation speed (A), mechanical milling time (B), powder-to-ball ratio (C), interaction effect of rotation speed with mechanical milling (AB), and the second-order terms of rotation speed (A) and powder-to-ball ratio (C). In particular, the R-square and R0square (adjusted) equal to 94.63% and 92.21%, respectively. These results suggest that the model is reliable at a confidence limit of 95%. Based on the ANOVA analysis, the regression model for the estimation of the UTS is given as follows.
Ultimate tensile strength (UTS) = 266.1 − 0.965A − 0.60B + 607C + 0.002598A^2^ − 6223C^2^ + 0.02407AB  in MPa,(3)

Table 6 shows the results of the ANOVA analysis based on the Box–Behnken design experiment for elongation, using the data presented in Table 2. It should be noted that the model is significant (*p* < 0.05). In addition, it appears that the rotation speed (A), mechanical milling time (B), powder-to-ball ratio (C), interaction effect of rotation speed with mechanical milling time (AB), and second order term of the mechanical milling time (B) have statistically significant effects on the elongation (*p* < 0.05). The R-square and R-square (adjusted) are shown to be 78.88% and 69.38%, respectively, which justifies the ANOVA model. The developed regression model for predicting elongation is presented as follows.
Elongation = −55.9 + 0.439A + 7.56B + 11C − 0.1687B^2^ − 0.01920AB + 19.09BC,(4)

As expected, the ANOVA results confirm that the rotation speed, mechanical milling time, and powder-to-ball ratio significantly affected the mechanical performance of the HS pure Cu. Interestingly, increasing the process parameter levels revealed that a higher process parameter levels caused the microstructural transformation in the material. Increasing the rotation speed, milling time, and powder-to-ball ratio allowed for more heavy, intense ball impacts on the powders during the milling process, which led to the formation of severely deformed layer and shell regions near the surface of the powders [61]. Such a change induced a higher degree of grain refinement, including the accumulation of defects. In particular, the reduction in grain size made dislocation movement more difficult because the neighboring grains had different crystal orientations; thus, it would need to change to a more appropriate orientation for dislocation to occur [62,63]. This realignment proved very difficult and an uncommon mechanism in the case of small grains. Moreover, it was found that small grain sizes resulted in more boundaries than when larger grain sizes were present. This characteristic also obstructed dislocation movement and resulted in a high UTS [64]. The correlation between grain size and tensile strength in HS pure Cu was also consistent with the Hall–Patch relationship [65]. Therefore, the UTS and elongation of the HS pure Cu were influenced by the microstructural characteristics, which depended on the magnitude of the process parameters in mechanical milling. Using the optimum set of process parameters would result in the appropriate microstructural characteristics and allow an excellent combination of UTS and elongation in the HS pure Cu.

### 3.3. Optimize Process Parameters Based on Mechanical Performance

The mechanical milling process parameters were optimized by applying the RSM, which was used to develop a numerical model, analyze the optimum set of process parameters, and validate the results [66]. To obtain the optimum set of process parameters for UTS and elongation, contour plots that demonstrated the possible independence of the various parameters were produced. To investigate the correlation between the process and response parameters, one parameter was set at the middle level and two parameters were mapped to the x and y axes, as shown in Figure 3a–c and Figure 4a–c. Such a contour plot was adequate for forecasting UTS and elongation for any region in the experimental area [67,68].

Figure 3a–c plot UTS as a function of mechanical milling time and rotation speed, powder-to-ball ratio and rotation speed, and powder-to-ball ratio and mechanical milling time, respectively. The UTS increased significantly with an increase in the milling time and rotation speed. Similar results were obtained by Corrochano et al. [59] when they investigated the fabrication of luminium matrix composites reinforced by ball milling. A maximum UTS of higher than 270 MPa is observed near the top right corner of these graphs. These results suggest that the recommend mechanical milling process parameters can be determined to be a rotation speed range of 180–200 rpm, mechanical milling time in the range of 17–20 h, and powder-to-ball-ratio in the range of 0.04–0.07. The simultaneous effect of mechanical milling time, rotation speed, and powder-to-ball ratio on the elongation are shown in Figure 4. The maximum elongation of higher than 45% is achieved at a rotation speed in the range of 190–200 rpm, mechanical milling time in the range of 16–18 h, and a powder-to-ball ratio in the 0.05–0.07 range. The increase in rotation speed resulted in an increase in elongation. Deng et al. [69] have indicated that the UST and elongation improved with increased milling time. Moreover, it should be mentioned that the range of possible process parameters needed for the best combination of both UTS and elongation overlapped. To determine the exact values of the optimum parameters, an RSM was conducted. According to the highest desirability function in the RSM analysis (0.8475), a rotation speed of 200 rpm, a mechanical milling time 17.78 h and a powder-to-ball ratio of 0.065 resulted in the best UST and elongation of 272.08 MPa and 46.85%, respectively in HS pure CU. The desirability was an objective function that ranged from zero to one, with a value of one being the most desirable combination. In this study, the maximum achievable desirability value is almost equal to one which led to the best combination of UTS and elongation.

### 3.4. Validation Adequacy of Numerical Model Using Optimized Values

In this study, the RSM was applied to investigate the optimal set of process parameters that influenced the best combination of UTS and elongation in HS pure Cu materials. To verify the accuracy of the developed optimal set of process parameters presented in the previous section, two more sets of samples were fabricated using the optimal process parameters, which were a rotation speed of 200 rpm, a mechanical milling time of 17.78 h and a powder-to-ball ratio of 0.065. However, the optimal condition of rotation speed was the maximum of the studied range (i.e., 200). The wider ranges of rotation speed, 220 and 240 rpm, have been forecasted through regression model for predicting UTS (Equation (3)) and elongation (Equation (4)). The predicted UTS were 284.72 and 299.49 MPa when rotation speed was 220 and 240 rpm, respectively. Although the predicted UTS with rotation speed of 220 and 240 rpm were increased, the predicted elongation was dropped to 46.67% and 46.21%, respectively. These predicted results demonstrate the trend that an increase in rotation speed results in UTS enhancement, while elongation was dropped. Sawangrat et al. [51] have indicated that the increase in rotation speed led to the increase in percentage of shell fraction. The results of the microstructural characteristics and mechanical properties are shown in Figure 5 and Table 7.

It has been demonstrated that both specimens showed identical mechanical behavior and shell volume fraction. The average UTS, elongation, and shell volume fractions were found to be 269.84, 44.37% and 44.75%, respectively. Interestingly, these experimental results are close to the predicted values, with average errors of 0.82% and 5.29% for UTS and elongation, respectively. These results agree with the RSM results and are satisfactory. Therefore, the regression model was adequate. The optimum set of UTS and elongation in HS pure Cu could be successfully prepared by applying a rotation speed of 200 rpm, a mechanical milling time of 17.78 h and a powder-to-ball ratio of 0.065.

### 3.5. Fracture Surface

To investigate the effect of the process parameters on the fracture behavior of the HS pure Cu, tensile specimens were prepared using the optimum set mechanical milling process parameters. Figure 6 shows a representative fracture surface of the HS pure Cu tensile specimens. It can be clearly observed that the fracture surface of the harmonic structure is dimpled, as shown in Figure 6a. According to Figure 6b, the larger surface dimples correspond to the core regions. On the other hand, the fine dimples correspond to the shell regions. Moreover, crack formation (de-bonding) occurs at the interface region between the shell and core. The reason why the cracks initiate at the core/shell interface is the different ductility between shell and core region. Therefore, the crack occurs more easily in the interface than in the center of the harmonic structure [17].

### 3.6. Fracture Mechanism

It is well-known that harmonic-structure materials exhibit uniform deformation behaviors due to the peculiar microstructural distribution of the shell and core regions [69,70]. The HS pure Cu also exhibited uniform deformation behavior because the shell and core regions were elongated to balance the shape change at the early stage of deformation, followed by cracks initiated at the shell and core interface regions [17]. Thus, the cracks are more easily initiated at the interface owing to the highly different strain concentration compared with the center of the shell and core regions [17]. In addition, cracks that developed from voids mostly occur at the shell and core interface regions. The cracks are effectively obstructed by the core regions, where the ductility is dominant [71]. However, the ductile core regions effectively impeded crack propagation and maintained strain, resulting in delayed plastic instability [18,58].

The complex fracture mechanism of HS pure Cu could be attributed to the specific topological distribution of the shell and core areas, which was influenced by the mechanical milling process parameters. The fracture mechanism model is shown in Figure 7. The fracture stage is described by four phases in the schematic. In the first phase of the fracture mechanism, the shell undergoes elongation along the tension direction to balance the specimen extension and elastically carries most of the loading [72]. Core regions are also elongated to balance the shape change and delay plastic deformation. It is found that the strength in the core regions increases due to strain hardening until it reaches the yield point, and the plastic deformation stage begins. Second, the strength of the shell region is comparable to the yield point, and the entire microstructure undergoes plastic deformation, followed by void initiation within both shell regions and interface regions between the shell and core [71]. Void nucleation occurs because of the difference in stress between the shell and core [72,73]. Third, the strain accumulation in both the core and shell regions continues with increasing strain. Cracks grow from voids in the shell regions, including shell and core interface regions, but are obstructed in the core regions due to their ductility [73], which in turn causes the microstructure to retain ductility. Finally, the strains accumulating in the core and shell regions are equal and do not maintain tension loading, which leads to cracks propagating the whole harmonic structure and the material fracturing [18,74,75]. It is clear that the shell and core regions play an important role in the fracture mechanism of the HS pure Cu. Void initiation and crack propagation also easily occur at the interface of the shell and core areas because of the difference in stress between these areas. This suggests that the controlled stress mismatch between the shell and core regions may have delayed void and crack generation, leading to delayed tensile fracture. In addition, the ductile core regions impede the propagation of cracks, resulting in enhanced ductility while retaining high strength in the HS pure Cu.

### 3.7. Influence Process Parameters on Fracture Mechanism

To determine the influence of the process parameters on the fracture mechanism in the HS pure Cu, an investigation into the degree of hardness variation between the core and shell areas was performed for the samples made using the three sets of processing conditions selected in Section 3.1. This information is plotted in Figure 8, which shows correlation in the hardness of the core and shell regions. The lowest degree of hardness variation was found in the minimum process parameter specimen, which was constructed using a rotation speed of 100 rpm, milling time of 10 h, and ball-to-powder ratio of 0.045. It appeared that continuously increasing process factor levels led to a higher degree of hardness variation in the core and shell regions, and induced the plastic instability that occurred in the early stage of the fracture mechanism, resulting in significantly reduced elongation, as shown in Table 5. This evidence suggests that increasing the process factor levels led to a higher degree of hardness variation between the core and shell areas and caused a reduction in the degree of adhesive bonding. Such a reduction allowed void and crack nucleation to appear more easily in the fracture mechanisms [72,73]. Likewise, the possibility of void spread and subsequent crack growth in HS pure Cu increased with increasing process parameter levels and governed the development of plastic instability in the early stage of deformation, resulting in low elongation. Hence, the mechanical properties of HS pure Cu could be improved by avoiding void and crack initiation in the core and shell interface regions, which could be successfully achieved by controlling the microstructural characteristics, especially the degree of hardness variation between the shell and core regions. It was found that creating an appropriate degree of hardness variation between the core and shell areas in the HS pure Cu depended on the selected process parameters, as presented in this paper. Therefore, our results indicate that the optimum set of process parameters for mechanical milling, which were a rotation speed of 200 rpm, mechanical milling time of 17.78 h, and powder-to-ball ratio of 0.065, allowed an appropriate degree of hardness variation between the core and shell regions. These results led to delayed void and crack initiation and extended plastic instability in the fracture mechanism, resulting in improvements in the UTS and elongation of up to 30% and 60%, respectively. Such improvements would reduce costs by not only reducing the usage of the material by providing high-strength small-sized components, but also allowing long shelf life of components owing to the best combination of UTS and elongation.

## 4. Conclusions

In this study, we systematically optimized the mechanical milling process parameters based on the mechanical properties and fracture mechanism of HS pure Cu by using the ANOVA and RSM. The following conclusions can be drawn:The consideration of the process parameters, rotation speed, mechanical milling time, and powder-to-ball ratio significantly influenced the UTS and elongation.Optimal process parameters, which were a rotation speed of 200 rpm, milling time of 17.78 ho, and powder-to-ball ratio of 0.065, allowed the best combination of UTS and elongation, which were 272.08 MPa and 46.85%, respectively.The HS pure Cu exhibited a very complex fracture mechanism. Superior strength and high maintained elongation were drawn from the shell and core regions, respectively. The voids and cracks mostly initiated at the core and shell interface regions owing to the strength mismatch between these regions.The increased process parameter values have an obvious effect on the fracture mechanism. These increased process parameters induced a large degree of strength variation between the core and shell areas, causing voids and cracks to appear more easily. Thus, the optimized process parameters allowed for control of the degree of strength variation and improved the mechanical performance of the HS pure Cu.

## Figures and Tables

**Figure 1 materials-15-08628-f001:**
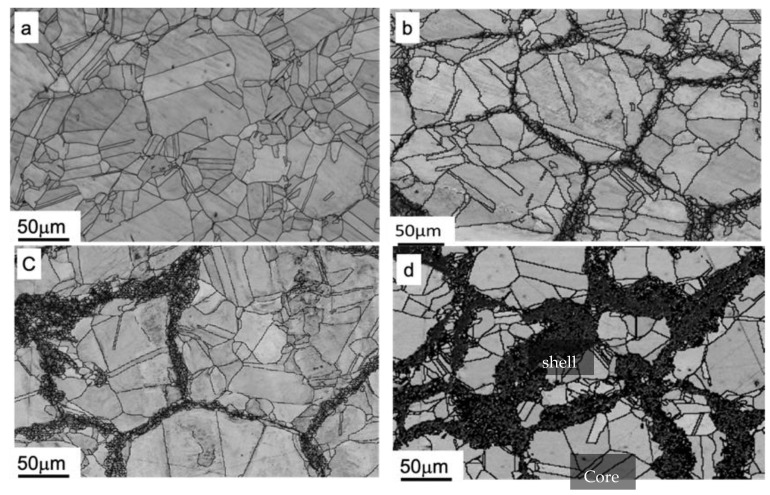
EBSD image quality maps overlaid with grain boundaries in cross section of compact prepared from (**a**) base material, (**b**) minimum process parameter levels (experiment #1), (**c**) medium process parameter levels (experiment #13), and (**d**) maximum process parameter levels (experiment #19).

**Figure 2 materials-15-08628-f002:**
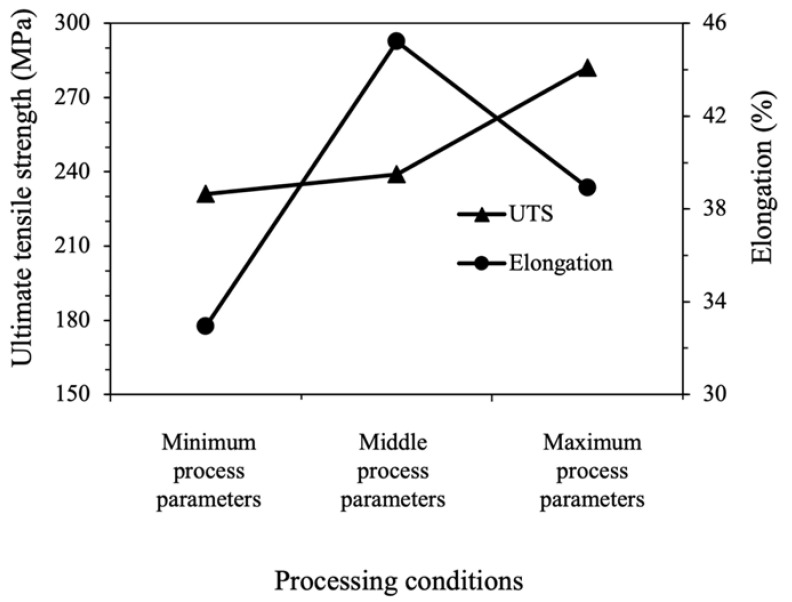
Correlation between UTS and elongation as a function of rotation speed (A) and mechanical milling time (B).

**Figure 3 materials-15-08628-f003:**
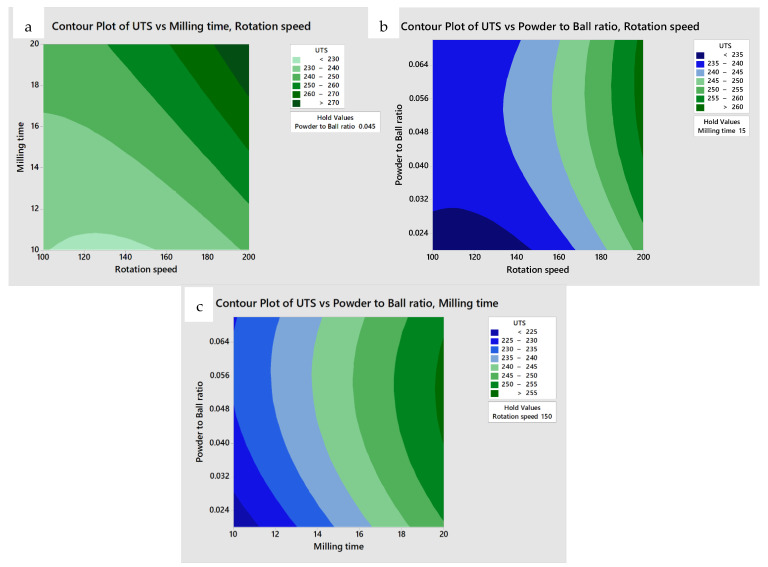
Contour plot of the effect of the (**a**) rotation speed and mechanical milling time, (**b**) rotation speed and powder-to-ball ratio, and (**c**) mechanical milling time and powder-to-ball ratio on UTS.

**Figure 4 materials-15-08628-f004:**
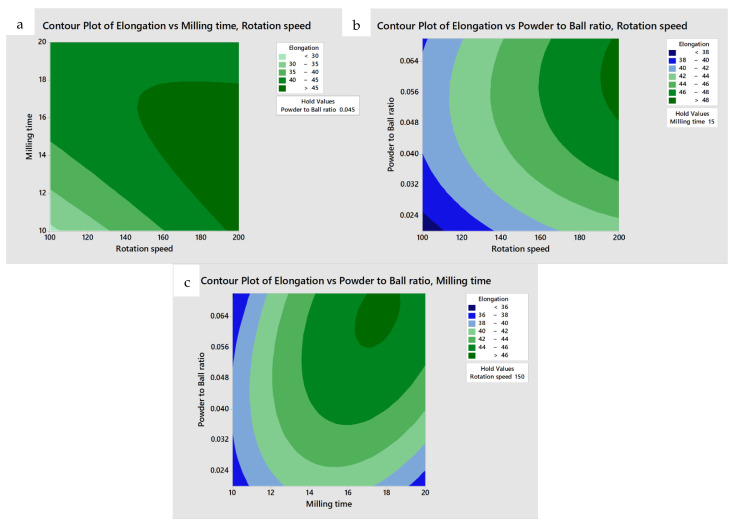
Contour plot of the effect of the (**a**) rotation speed and mechanical milling time, (**b**) rotation speed and powder-to-ball ratio, and (**c**) mechanical milling time and powder-to-ball ratio on elongation.

**Figure 5 materials-15-08628-f005:**
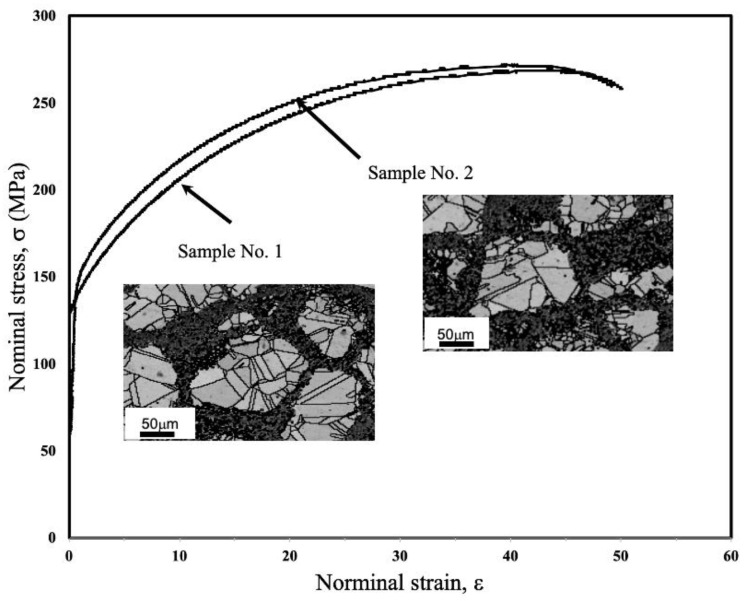
Nominal stress–nominal strain curves in pure Cu prepared using the optimum set of mechanical milling process parameters.

**Figure 6 materials-15-08628-f006:**
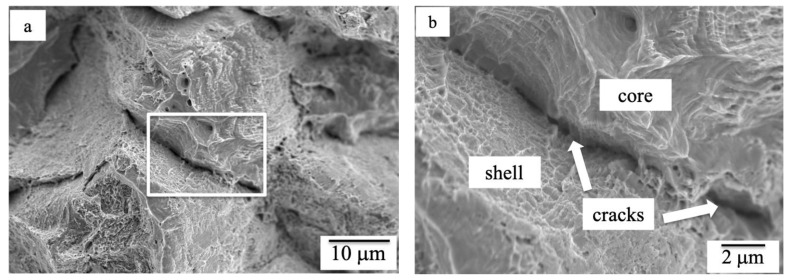
SEM micrograph of fracture surface of HS pure Cu after tensile testing: (**a**) low magnification and (**b**) enlarge of rectangular in (**a**).

**Figure 7 materials-15-08628-f007:**
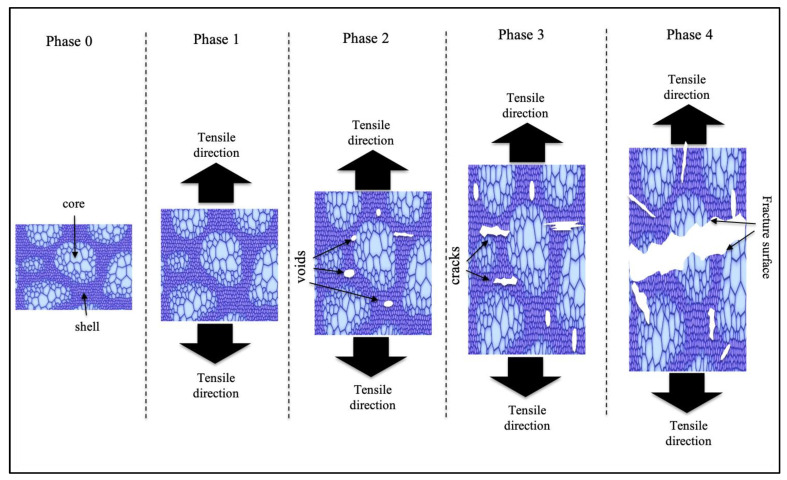
Fracture mechanism model of HS pure Cu.

**Figure 8 materials-15-08628-f008:**
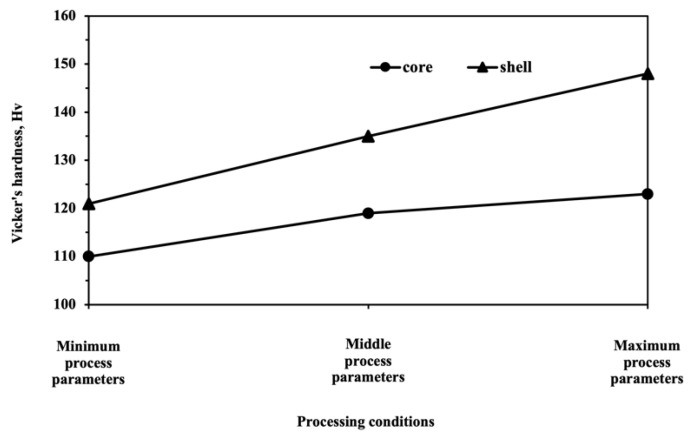
Plot of the hardness for the core and shell regions as a function of process parameters.

**Table 1 materials-15-08628-t001:** Range of each process parameter in the mechanical milling process in this study.

Parameters	Notation	Units	Level		
			−1	0	1
Rotation speed	A	rpm	100	150	200
Mechanical milling time	B	hours	10	15	20
Powder-to-ball ratio	C	–	0.02	0.045	0.07

**Table 2 materials-15-08628-t002:** The design of the experiment using Box-Behnken design.

Exp. No.	Run No.	Coded Level			Original Value		
		A	B	C	A	B	C
1	25	−1	−1	0	100	10	0.045
2	11	1	−1	0	200	10	0.045
3	13	−1	1	0	100	20	0.045
4	14	1	1	0	200	20	0.045
5	19	−1	0	−1	100	15	0.02
6	27	1	0	−1	200	15	0.02
7	26	−1	0	1	100	15	0.07
8	6	1	0	1	200	15	0.07
9	1	0	−1	−1	150	10	0.02
10	3	0	1	−1	150	20	0.02
11	30	0	−1	1	150	10	0.07
12	8	0	1	1	150	20	0.07
13	20	0	0	0	150	15	0.045
14	23	0	0	0	150	15	0.045
15	10	0	0	0	150	15	0.045
16	7	−1	−1	0	100	10	0.045
17	29	1	−1	0	200	10	0.045
18	12	−1	1	0	100	20	0.045
19	18	1	1	0	200	20	0.045
20	28	−1	0	−1	100	15	0.02
21	24	1	0	−1	200	15	0.02
22	22	−1	0	1	100	15	0.07
23	9	1	0	1	200	15	0.07
24	16	0	−1	−1	150	10	0.02
25	17	0	1	−1	150	20	0.02
26	4	0	−1	1	150	10	0.07
27	2	0	1	1	150	20	0.07
28	21	0	0	0	150	15	0.05
29	5	0	0	0	150	15	0.045
30	15	0	0	0	150	15	0.045

**Table 3 materials-15-08628-t003:** Microstructural characteristics and mechanical properties of pure Cu specimens.

Materials	Mean Grain Size (μm)		UTS (MPa)	Elongation (%)	Hardness (Hv)		Shell Fraction (%)
	Core	Shell			Core	Shell	
Base material	33.4 ± 2.07	–	209.16	29.3	114 ± 4.13	–	–
Minimum process parameter levels	30.12 ± 1.46	4.3 ± 1.22	231.03	32.94	110 ± 2.42	121 ± 0.89	12.42
Medium process parameter levels	28.62 ± 1.05	3.6 ± 0.75	239.1	45.22	119 ± 2.10	135 ± 1.39	20.14
Maximum process parameter levels	25.87 ± 1.69	1.8 ± 1.21	282.29	38.93	123 ± 1.82	148 ± 2.67	42.84

**Table 4 materials-15-08628-t004:** The results of UTS and elongation according to the Box–Behnken design.

Exp. No.	UTS (MPa)	Elongation (%)	Exp. No.	UTS (MPa)	Elongation (%)
1	231.03	32.943	16	234.43	31.03
2	244	45.82	17	242	43.92
3	245	42.93	18	243	46.92
4	274.4	38.29	19	282.29	38.93
5	235	38.93	20	235	35.93
6	258	46.2	21	254	47.9
7	229.52	35.93	22	232.73	33.91
8	260.21	49.26	23	257.23	47.93
9	214.94	33.92	24	219	33.09
10	245	34.02	25	249	36.92
11	230.32	37.93	26	232.63	35.03
12	260.32	49.72	27	258	48.2
13	245.92	44.92	28	244.32	45.92
14	243.02	45.02	29	243.8	45.3
15	242.03	46.39	30	239.43	43.29

**Table 5 materials-15-08628-t005:** Results of ANOVA performed on the ultimate tensile strength results.

Source	Sum of Squares	df	Mean Square	F-Value	*p*-Value
Model	5858.62	9	650.96	39.13	0.000
A	2075.08	1	2075.08	124.73	0.000
B	2703.63	1	2703.63	162.52	0.000
C	162.69	1	162.69	9.78	0.000
AB	289.80	1	289.80	17.42	0.000
AC	18.82	1	18.82	1.13	0.300
BC	5.85	1	5.85	0.35	0.560
A^2^	311.62	1	311.62	18.73	0.000
B^2^	0.03	1	0.03	0.00	0.966
C^2^	101.08	1	101.08	6.08	0.023
Lack of fit	228.07	3	76.02	12.35	0.000
Pure error	104.65	17	6.16		
Total	6191.34	29			
	R-sq	94.63%	R-sq(adj)	92.21%	

**Table 6 materials-15-08628-t006:** Results of ANOVA performed on the elongation results.

Source	Sum of Squares	df	Mean Square	F-Value	*p*-Value
Model	790.01	9	87.779	8.30	0.000
A	211.88	1	211.88	20.04	0.000
B	90.52	1	90.52	8.56	0.008
C	60.06	1	60.06	5.68	0.027
AB	184.29	1	184.29	17.43	0.000
AC	2.78	1	2.78	0.26	0.614
BC	46.49	1	46.49	4.40	0.049
A^2^	5.01	1	5.01	0.47	0.499
B^2^	131.42	1	131.42	12.43	0.002
C^2^	27.01	1	27.01	2.55	0.126
Lack of fit	175.21	3	58.40	27.38	0.000
Pure error	104.65	29	1001.48		
Total	6191.34	29			
	R-sq	78.88%	R-sq(adj)	69.38%	

**Table 7 materials-15-08628-t007:** Microstructural characteristics and mechanical properties of pure Cu specimens prepared by optimum set of process parameters in mechanical milling.

Sample No.	Experiment Result		Shell Volume Fraction (%)	Predicted		Error (%)	
	UTS (MPa)	Elongation (%)		UTS (MPa)	Elongation (%)	UTS	Elongation
1	268.51	45.27	43.19	272.08	46.85	1.31	3.37
2	271.18	43.47	46.32			0.33	7.21
Average	269.84	44.37	44.75			0.82	5.29
STD.	1.89	1.27					

## Data Availability

Data are contained within the article.

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
