# Peer review of "Fabrication of Copper of Harmonic Structure: Mechanical Property-Based Optimization of the Milling Parameters and Fracture Mechanism"

_materials, 2022, doi:10.3390/ma15238628_

Round 1
Reviewer 1 Report
This work aims to construct harmonic structured pure Cu by optimizing the process parameters of mechanical milling process and revealing the fracture mechanism. However, there are some mistakes need to be revised.
1. The prime novelty of the manuscript could be enhanced by using more scientific terminology. The language should be polished.
2. Please check the number of keywords. Is there any quantitative limit?
3. The contents in 3.1 (Box–Behnken design results) seems more about experimental method introduction. The author needs to consider whether it is proper to put this part in discussion.
4. The optimum experimental parameters of rotation speed and power-to-ball ratio are 200 rpm, 0.065, respectively according to RSM. However, maximum range of calculation is 200 rpm, wider range of forecast results is more reasonable to be provided, like 220 or 240 rpm.
Author Response
"Please see the attachment"

Reviewer 2 Report
It is a great article - my only remark concerns with the measured result values in the various tables - for those please support the accuracy of the values within the measurement error range.
Besides of that, this is an excellent article
Author Response
"Please see the attachment."

Reviewer 3 Report
The manuscript is suitable for this journal, it is clear and presented in a well-structured form, supplemeted by the photos, figures, tables. The topic of the manuscript is interesting and scientifically appropriate.
Figures are related with the text and data.
References are not appropriate (mainly in the section Introduction), and so it must to be repaired. The manuscript is devoted to the issue (mechanical milling, powder metallurgy, statistical and numerical models,...) that is being researched in research centers around the world. The citations in the article are limited to only one part of the world, which is why it is necessary to cite works from other parts of the world (such as Europe and USA), where important knowledge in the studied issue was created. It is necessary to add more suitable citations to the text in some cases.
The presented model and results of the manuscript are reproducible on the basis of the parts mentioned in the manuscript, but some details is necessary to fill (see specific comments).
Conclusions are consistended with argumets in the manuscript.
Specific comments:
1. The term „three levels“ (line 131, Table 1, Table 2) needs to be explained in more details in the text of manuscript.
2. Try to specify in more detailes differences between core and shell in the text of manuscript (no only in figure).
3. UTS and Elongationa are functions of parametres A, B, and C. But how exactly were deteminated values of UTS and Elongation in Table 2 (equation, model, program...)?
References are not appropriate (mainly in the section Introduction), and so it must to be repaired.

Round 2
Reviewer 3 Report
I am satisfied with the answers, thank you.
Author Response
Dear reviewer,
Thank you for your reply.